# Machinability enhancement of heat-treated Incoloy 800H during turning using laser-textured carbide inserts under semi-solid MoS₂ lubrication

**Palanisamy Angappan[1], Palanisamy Duraisamy[2], Prakash Chellamuthu[3], Abhishek Agarwal [4]\*, Lenin Kasirajan[5]**

**1** Department of Mechanical Engineering, Surya Engineering College, Erode, Tamil Nadu, India, **2** Department of Mechanical Engineering, Adhi College of Engineering and Technology, Kancheepuram, Tamil Nadu, India, **3** Department of Mechanical Engineering, Roever Engineering College, Perambalur, Tamil Nadu, India, **4** Department of Mechanical Engineering, College of Science and Technology, Royal University of Bhutan, Phuentsholing, Bhutan, **5** Department of Mechanical Engineering, K. Ramakrishnan College of Engineering, Tiruchirappalli, Tamil Nadu, India

\* agarwala.cst@rub.edu.bt

## Abstract

Incoloy 800H is an iron–nickel–chromium based superalloy widely used in high-temperature applications, but its poor machinability due to rapid work hardening and low thermal conductivity makes machining difficult. This study experimentally investigates the machinability of heat-treated Incoloy 800H during CNC turning using laser-textured tungsten carbide cutting inserts combined with semi-solid MoS₂ lubrication. Four tool configurations were examined: non-textured (NT), parallel-textured (PT), elliptical-textured (ET), and semi-elliptical-textured (SCT) inserts. Turning experiments were performed by varying cutting speed (35–55 m/min) and feed rate (0.02–0.06 mm/rev) while maintaining a constant depth of cut of 1 mm. Machining performance was evaluated through surface roughness, cutting force, cutting power, specific energy consumption, specific cutting pressure, tool-tip temperature, and material removal rate. The results indicate that the parallel-textured tool exhibited superior machining performance compared with the other inserts. The lowest surface roughness of 0.30 µm, cutting force of 91.23 N, and tool-tip temperature of 73.2 °C were obtained with the PT insert under optimal cutting conditions. The improved performance is attributed to reduced tool–chip contact length and enhanced lubrication at the interface provided by the textured surface combined with MoS₂ lubrication. Scanning electron microscopy and energy-dispersive X-ray analysis confirmed diffusion and abrasion as the dominant tool wear mechanisms. The findings demonstrate that the combined application of laser surface texturing and semi-solid MoS₂ lubrication significantly improves the machinability of Incoloy 800H during turning operations. The study provides experimental evidence that parallel-oriented surface textures effectively reduce tool–chip contact length and frictional interaction, thereby enhancing machining performance for difficult-to-cut superalloys.

**Data availability statement:** All relevant data supporting the findings of this study are available within the manuscript and its Supporting information file S1 Data.

**Funding:** The author(s) received no specific funding for this work.

**Competing interests:** The authors have declared that no competing interests exist.

## Introduction

Turning is one of the most widely used machining operations in manufacturing industries [1]. The machining performance during turning is strongly influenced by cutting parameters such as cutting speed, feed rate, depth of cut, tool geometry, and the properties of the work material [2]. Nickel-based and iron-nickel superalloys are widely used in high-temperature environments because of their excellent mechanical strength, oxidation resistance, and corrosion resistance [3,4]. However, these alloys are classified as difficult-to-machine materials due to their low thermal conductivity, rapid work hardening, and high chemical affinity with cutting tool materials [5–7].

Incoloy 800H is an iron-nickel-chromium based superalloy commonly used in components exposed to high temperature conditions such as heat exchangers, nuclear reactors, petrochemical equipment, steam generator tubes, and pressure vessels [8–10]. These applications often involve service temperatures ranging from 550 °C to 700 °C [3,4]. Although Incoloy 800H exhibits excellent mechanical and thermal properties at elevated temperatures, its machining performance is relatively poor because of severe work hardening and high cutting temperatures during machining operations [11,12].

During machining of superalloys, cutting forces are typically higher than those required for conventional alloy steels. The low thermal conductivity of these materials leads to higher temperatures at the tool–chip interface, which accelerates tool wear and reduces tool life [5,6]. Surface integrity of machined components, including surface roughness and residual stress, is significantly influenced by machining conditions and tool characteristics [13,14]. Therefore, improving machinability through optimized cutting tools and lubrication techniques has become an important research focus [15,16].

Several studies have investigated the influence of cutting parameters and tool materials on the machining performance of nickel-based and iron-based superalloys [17–19]. Advanced lubrication strategies such as minimum quantity lubrication, hybrid cooling techniques, and solid lubrication have also been explored to reduce friction and cutting temperature during machining of nickel-based superalloys [20–23]. Among these techniques, surface texturing of cutting tools has attracted considerable attention because it can reduce tool–chip contact area and enhance lubrication retention at the interface [24–26]. Recent studies have demonstrated that hybrid lubrication strategies such as nano-enhanced minimum quantity lubrication, cryogenic-assisted machining, and laser-textured cutting tools significantly improve machinability of nickel-based and iron–nickel superalloys by reducing tool–chip friction and cutting temperature [27–32]. In particular, the synergistic interaction between micro-scale surface textures and solid lubricant retention has been reported to enhance tool life and surface integrity during turning of difficult-to-machine alloys [33–35].

Laser surface texturing has emerged as an effective technique for modifying tool surfaces and improving tribological performance during machining operations [36,37]. Micro-textures fabricated on the rake face of cutting tools can act as reservoirs for lubricants and reduce friction at the tool–chip interface, thereby improving cutting

performance and reducing tool wear [22,38,39]. Despite the availability of several studies on textured cutting tools and advanced lubrication strategies, limited experimental investigations have examined the combined influence of laser surface texture geometry and semi-solid $MoS_2$ lubrication on the machinability of heat-treated Incoloy 800H. In particular, the role of texture orientation relative to chip flow direction and its interaction with lubricant retention behaviour has not been systematically evaluated. Therefore, the objective of the present study is to investigate the influence of parallel, elliptical, and semi-elliptical laser-textured carbide inserts under semi-solid $MoS_2$ lubrication on machining performance during CNC turning of heat-treated Incoloy 800H. The novelty of this work lies in the combined evaluation of texture geometry, semi-solid lubrication, and heat-treated microstructural condition on multiple machinability indicators including surface roughness, cutting force, tool-tip temperature, specific energy consumption, specific cutting pressure, and tool–chip contact length.

## Materials and methods

Forged rods of Incoloy 800H superalloy with a diameter of 30 mm and length of 120 mm were selected based on standard cylindrical turning practice to ensure sufficient machining length for steady-state cutting conditions and repeatability of experimental measurements. Similar specimen dimensions have been adopted in previous machinability investigations on nickel-based superalloys [26,30,31]. The as-received specimens were subjected to a solution heat treatment process in order to obtain a stable microstructure representative of industrial high-temperature components. The rods were heated at a rate of 40 °C/min to a temperature of 1075 °C and held for 1 h in an electric induction furnace with a maximum operating capacity of 1300 °C, followed by water quenching. Subsequently, the specimens were aged at 975 °C for 1 h using the same heating rate and then furnace cooled to room temperature. The chemical composition of Incoloy 800H was determined through spectroscopy analysis at the Test Point research laboratory, Coimbatore, India, and the measured values (Table 1) were found to be consistent with the standard compositional limits specified for Alloy 800H in ASTM B407 and previous literature reports [9,10]. The heat-treated specimens were used for the turning experiments. This heat-treatment condition was selected to simulate the microstructural state commonly encountered in industrial components operating under high-temperature service conditions [40]. Although such treatment improves thermal stability and mechanical strength, it also increases machining difficulty due to enhanced work-hardening behaviour and reduced thermal conductivity [41]. Turning experiments were performed on a LEADWELL T5 CNC turning centre equipped with a FANUC control system (maximum spindle speed 4500 rpm, power 7.5 kW). The experimental setup used for measuring cutting forces is shown in Fig 1. The machining parameters considered in this study were cutting speed (Vc), feed rate (f), and depth of cut (ap). ISO grade K20 tungsten carbide (WC) uncoated cutting inserts (CNMA 120408-THM, WIDIA, India) were used during the experiments. Tungsten carbide inserts are widely employed for machining nickel-based superalloys due to their high hardness, wear resistance, and thermal stability under elevated cutting temperatures [19]. The tool geometry specifications were: clearance angle 5°, side rake angle −6°, inclination angle −6°, approach angle 95°, point angle 80°, and nose radius 0.8 mm. Laser surface textures were fabricated on the rake face of the cutting inserts using a Nd:YAG solid-state laser system (Model LDP 400 MQ, LEE Laser Inc., USA).

The texturing process was carried out at Meeras Laser Solutions, Chennai, India. Laser surface texturing was selected because it allows precise fabrication of micro-scale texture geometries without significantly altering the bulk mechanical properties of the cutting insert. The selected texture geometries (parallel, elliptical, and semi-elliptical) were chosen to examine the influence of texture orientation relative to chip flow direction, which significantly affects lubricant retention

**Table 1. Chemical composition of Incoloy 800H (wt %) determined by spectroscopy analysis and verified against standard Alloy 800H composition ranges reported in the literature [9,10].**

| C | Mn | P | S | Si | Cr | Ni | Ti | Al | Cu | Fe |
|---|---|---|---|---|---|---|---|---|---|---|
| 0.069 | 0.76 | 0.014 | 0.001 | 0.13 | 20.42 | 31.59 | 0.57 | 0.50 | 0.42 | 45.562 |

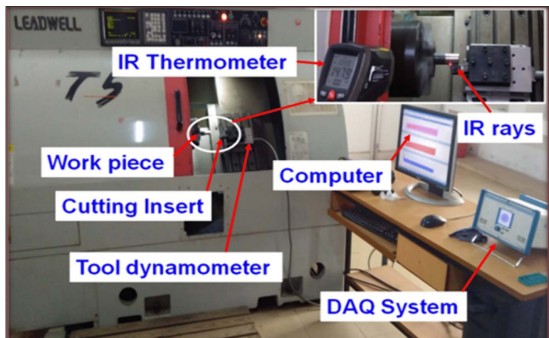

**Fig 1. Experimental setup of the CNC turning centre showing the dynamometer-based cutting force measurement system.**

and friction behaviour during machining. The generated micro-textures act as lubricant reservoirs and reduce the effective tool–chip contact area during machining, thereby improving tribological performance at the cutting interface. Semi-solid $MoS_2$ lubrication was applied externally along the tool–chip interface prior to machining using a thin and uniform coating layer. $MoS_2$ was selected because of its lamellar crystal structure, which provides low shear strength between sliding layers and enables effective solid lubrication under boundary lubrication conditions typically encountered during machining of nickel-based superalloys. The semi-solid $MoS_2$ layer formed a stable lubricating tribofilm at the tool–chip interface, reducing adhesion, frictional resistance, and localized temperature rise during machining of the heat-resistant alloy. The fabricated laser surface textures had an average groove width of 120 μm, depth of 35 μm, and spacing of 250 μm, and were positioned at an offset distance of approximately 150 μm from the cutting edge to prevent weakening of the cutting edge while ensuring effective interaction with the tool–chip contact region. The selected texture depth (35 μm) remained greater than the observed crater wear depth during the experimental cutting duration, ensuring that the functional role of the micro-textures in lubricant retention and friction reduction was maintained throughout the machining trials. Laser texturing was performed using a pulsed Nd:YAG laser operating at a wavelength of 1064 nm, with an average power of 18 W, pulse frequency of 20 kHz, and scanning speed of 300 mm/s under controlled pulse energy and scanning overlap conditions to ensure uniform geometry and repeatability of the fabricated micro-textures without inducing thermal damage near the cutting edge. The selected texture geometry and spacing were designed to enhance lubricant retention while preserving cutting-edge strength and maintaining the mechanical integrity of the insert during machining operations. Three textured tools namely Parallel Textured tool (PT), Elliptical Textured tool (ET) & Semi Elliptical Textured tool (SCT) were used and obtained results were compared with non-textured tool (NT). An ISO labeled PCLNL 1610 M12 tool holder was used, it is mounted on a three-component piezoelectric dynamometer (KISTLER 9257B) allowing measurements from 5 to 5 kN.

The force components measurement-chain comprises a charge amplifier (KISTLER 5019B130) with data acquisition hardware unit (A/D 2855A3) and the graphical programming environment (DYNOWARE 2825A1-1) for the data analysis and visualization. During cutting force analysis, the average value of the force peaks in the steady-state cutting region was considered for evaluation in order to minimize the influence of transient effects during tool entry and exit [42]. The force signals were acquired at a sampling frequency of 5000 Hz. All force components were recorded continuously during machining and processed using the DYNOWARE data acquisition software. The force signals gathered were analyzed for a cutting time of 5s. Each machining experiment was repeated three times under identical cutting conditions, and the reported values represent the average of the measured responses in order to improve measurement reliability and reduce experimental uncertainty. The standard deviation of repeated measurements remained within ±4% of the mean values, indicating good repeatability of the experimental observations and confirming that measurement uncertainty had negligible influence on the reported machinability trends. The machined samples surface roughness were measured using surface

roughness tester (Mitutoyo, SJ301) in four different locations, with cut off length 0.8 mm and traverse length 5 mm and the average values are reported. The cutting tool-tip temperature was measured at the rake-face region near the tool–chip interface using a calibrated infrared thermometer (HTC Instruments IRX 68; distance-to-spot ratio 50:1, dual-laser targeting, response time 150 ms, measurement capacity up to 1850 °C) during turning operations. Two measurements were recorded at each cutting condition and the average value was reported to minimize measurement uncertainty. This study does not involve human participants or animals; therefore, ethical approval was not required. The optical microstructure and SEM images of the heat-treated Incoloy 800H samples are shown in Fig 2a–d. The microstructures of the laser-textured cutting inserts are presented in Fig 3a non-textured (NT), Fig 3b parallel-textured (PT), Fig 3c elliptical-textured (ET), and Fig 3d semi-elliptical-textured (SCT) inserts.

Machining power was calculated using the following relation [43]:

$$P = Fz \times Vc \tag{1}$$

where P is the cutting power (W), Fz is the main cutting force (N), and Vc is the cutting speed (m/s). The cutting speed used in Eq. (1) was converted from m/min to m/s for the calculation of cutting power.

Specific energy consumption (SEC) represents the energy required to remove a unit volume of material and is expressed as [19]:

$$SEC = P/MRR \tag{2}$$

The specific cutting pressure (SCPR) was calculated using [44]:

$$SCPR = Fz/A \tag{3}$$

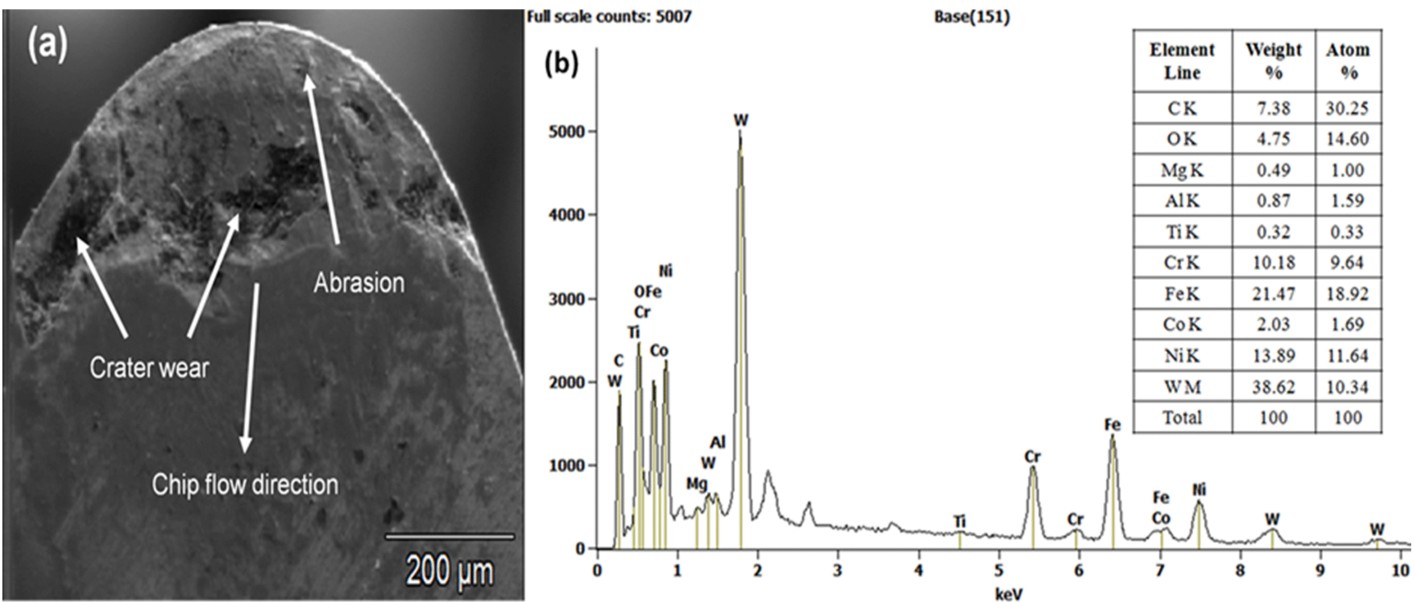

**Fig 2. Optical microstructure and SEM images of heat-treated Incoloy 800H work material.**

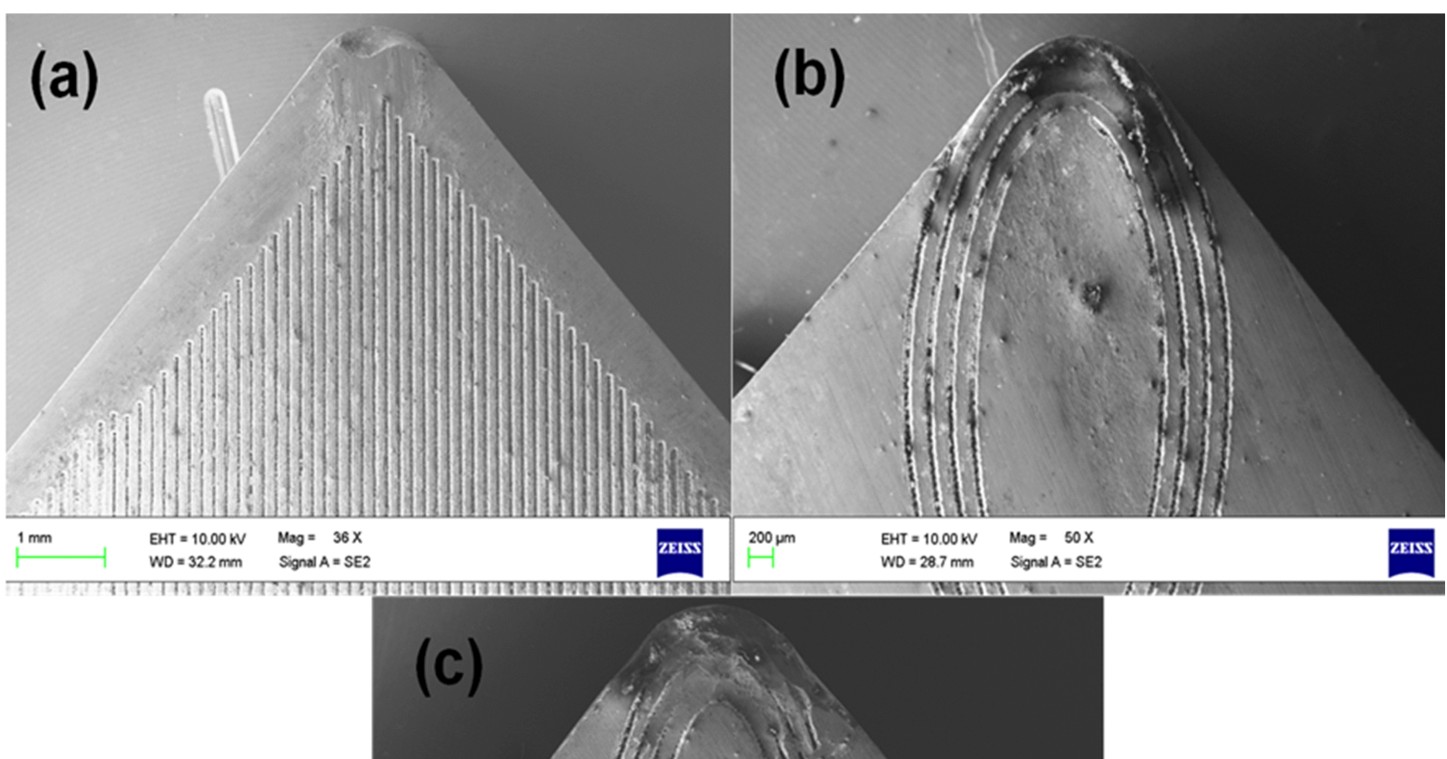

**Fig 3. Laser-textured carbide inserts used in the experiments: (a) non-textured (NT), (b) parallel textured (PT), (c) elliptical textured (ET), and (d) semi-elliptical textured (SCT).**

where Fz is the cutting force and A is the uncut chip cross-sectional area (f × ap). Material removal rate (MRR) [45] during turning was determined using:

$$MRR = Vc \times f \times ap \tag{4}$$

where Vc is the cutting speed (m/min), f is the feed rate (mm/rev), and ap is the depth of cut (mm). The machining parameters & their experimental design are shown in Table 2.

Table 2 presents the experimental design consisting of three levels of cutting speed and feed rate with a constant depth of cut. A full factorial experimental design (3 × 3) was employed to systematically evaluate the individual and combined

**Table 2. Experimental design layout for the turning experiments indicating cutting speed (Vc), feed rate (f), and depth of cut (ap).**

| Ex. No. | Cutting speed(Vc), (m/min) | Feed rate (f), (mm/rev) | Depth of cut (ap), (mm) |
|---|---|---|---|
| 1 | 35 | 0.02 | 1 |
| 2 | 35 | 0.04 | 1 |
| 3 | 35 | 0.06 | 1 |
| 4 | 45 | 0.02 | 1 |
| 5 | 45 | 0.04 | 1 |
| 6 | 45 | 0.06 | 1 |
| 7 | 55 | 0.02 | 1 |
| 8 | 55 | 0.04 | 1 |
| 9 | 55 | 0.06 | 1 |

influence of machining parameters and tool surface textures on machinability performance. The selected parameter ranges were chosen based on recommended machining conditions for nickel-based superalloys reported in previous studies [17–19,46], representing practical low- and moderate-speed turning regimes commonly adopted for difficult-to-machine alloys. The depth of cut was maintained constant at 1 mm, which lies within the stable machining range recommended for carbide inserts during turning of heat-resistant superalloys, in order to isolate the influence of cutting speed and feed rate without introducing additional variability in cutting load. Statistical evaluation of the experimental results was performed using analysis of variance (ANOVA) at a 95% confidence level to assess the significance of machining parameters on the measured responses. Feed rate showed the strongest influence on surface roughness, whereas cutting speed significantly affected cutting force and tool-tip temperature. Interaction effects between cutting speed and feed rate were significant for specific energy consumption and specific cutting pressure. A summary of the statistical significance is presented in Table 3. The raw experimental dataset used for statistical evaluation and generation of all plots is provided in Supporting Information file S1 Data.

The ANOVA results confirm that feed rate predominantly influences surface roughness, whereas cutting speed significantly affects cutting force and tool-tip temperature, while interaction effects govern specific energy consumption and specific cutting pressure.

## Results and discussion

### Effect of surface textures on the surface roughness

The measured machining responses obtained during the experiments are summarized in Tables 4 and 5. Surface roughness, cutting force, and cutting power values are listed in Table 4, while specific energy consumption, specific cutting

**Table 3. Summary of ANOVA results showing significance of machining parameters on machining responses (95% confidence level).**

| Response parameter | Most significant factor | Secondary factor | Interaction effect (Vc×f) |
|---|---|---|---|
| Surface roughness (Ra) | Feed rate | Cutting speed | Moderate |
| Cutting force (Fz) | Cutting speed | Feed rate | Low |
| Cutting power (P) | Cutting speed | Feed rate | Low |
| Specific energy consumption (SEC) | Interaction | Feed rate | Significant |
| Specific cutting pressure (SCPR) | Interaction | Cutting speed | Significant |
| Tool-tip temperature (TT) | Cutting speed | Feed rate | Moderate |

**Table 4. Experimental results of surface roughness, cutting force, cutting power for various textured tools.**

| Ex. No. | Surface roughness (R$_a$), µm | | | | Cutting force (Fz), N | | | | Cutting power(P), W | | | |
|---|---|---|---|---|---|---|---|---|---|---|---|---|
| | NT | PT | ET | SCT | NT | PT | ET | SCT | NT | PT | ET | SCT |
| 1 | 0.67 | 0.54 | 0.77 | 1.09 | 161.62 | 142.36 | 151.23 | 188.26 | 94.28 | 83.04 | 88.22 | 109.82 |
| 2 | 0.78 | 0.74 | 0.98 | 1.03 | 169.62 | 148.75 | 156.24 | 195.39 | 98.95 | 86.77 | 91.14 | 113.98 |
| 3 | 1.03 | 1.00 | 1.23 | 1.25 | 201.29 | 187.14 | 192.34 | 210.67 | 117.42 | 109.17 | 112.20 | 122.89 |
| 4 | 0.44 | 0.40 | 0.62 | 0.85 | 124.08 | 122.56 | 130.79 | 136.71 | 93.06 | 91.92 | 98.09 | 102.53 |
| 5 | 0.58 | 0.41 | 0.75 | 0.94 | 151.92 | 139.73 | 159.41 | 166.45 | 113.94 | 104.80 | 119.56 | 124.84 |
| 6 | 0.45 | 0.44 | 0.83 | 1.02 | 183.78 | 158.79 | 164.86 | 197.43 | 137.84 | 119.09 | 123.65 | 148.07 |
| 7 | 0.31 | 0.30 | 0.67 | 1.01 | 101.38 | 91.23 | 102.16 | 112.49 | 92.93 | 83.63 | 93.65 | 103.12 |
| 8 | 0.34 | 0.32 | 0.45 | 0.86 | 120.30 | 111.45 | 114.68 | 134.27 | 110.28 | 102.16 | 105.12 | 123.08 |
| 9 | 0.36 | 0.34 | 0.51 | 1.03 | 134.80 | 120.14 | 129.47 | 141.68 | 123.57 | 110.13 | 118.68 | 129.87 |

**Table 5. Experimental results of Specific energy consumption, Specific cutting pressure, Tool-tip temperature & MRR for various textured tools.**

| Ex. No. | Specific energy consumption (SEC), J/mm³ | | | | Specific cutting pressure (SCPR), N/mm² | | | | Tool-tip temperature (TT),˚C | | | | MRR,m-m³/s |
|---|---|---|---|---|---|---|---|---|---|---|---|---|---|
| | NT | PT | ET | SCT | NT | PT | ET | SCT | NT | PT | ET | SCT | |
| 1 | 16.17 | 14.24 | 15.13 | 18.84 | 16162 | 14236 | 15123 | 18826 | 88.6 | 75.7 | 80.2 | 92.4 | 5.83 |
| 2 | 8.48 | 7.44 | 7.81 | 9.77 | 8481 | 7437.5 | 7812 | 9769.5 | 92.7 | 78.1 | 83.7 | 96.8 | 11.67 |
| 3 | 6.71 | 6.24 | 6.41 | 7.02 | 6709.6 | 6238 | 6411.3 | 7022.3 | 95.4 | 81.5 | 87.3 | 100.8 | 17.50 |
| 4 | 12.41 | 12.26 | 13.08 | 13.67 | 12408 | 12256 | 13079 | 13671 | 119.5 | 97.3 | 108.6 | 112.8 | 7.50 |
| 5 | 7.60 | 6.99 | 7.97 | 8.32 | 7596 | 6986.5 | 7970.5 | 8322.5 | 123.3 | 105.9 | 117.9 | 121.9 | 15.0 |
| 6 | 6.13 | 5.29 | 5.50 | 6.58 | 6126 | 5293 | 5495.3 | 6581 | 125.3 | 73.2 | 81.7 | 88.6 | 22.50 |
| 7 | 10.13 | 9.12 | 10.21 | 11.25 | 10138 | 9123 | 10216 | 11249 | 88.3 | 78.2 | 86.7 | 93.6 | 9.17 |
| 8 | 6.02 | 5.57 | 5.73 | 6.71 | 6015 | 5572.5 | 5734 | 6713.5 | 113.6 | 92.3 | 97.6 | 101.2 | 18.33 |
| 9 | 4.49 | 4.00 | 4.32 | 4.72 | 4493.3 | 4004.6 | 4315.6 | 4722.6 | 121.6 | 99.7 | 111.3 | 116.9 | 27.50 |

pressure, tool-tip temperature, and material removal rate are presented in Table 5. The experimental results indicate that the surface roughness values ranged from 0.31–1.03 µm for the non-textured tool (NT), 0.30–1.00 µm for the parallel textured tool (PT), 0.45–1.23 µm for the elliptical textured tool (ET), and 0.85–1.25 µm for the semi-elliptical textured tool (SCT). This trend is consistent with the ANOVA results presented in Table 3, which identify feed rate as the most significant factor influencing surface roughness at the 95% confidence level. Among the tested tools, the PT insert produced the lowest surface roughness value of 0.30 µm. This improvement can be attributed to the orientation of the parallel micro-textures, which promote lubricant retention at the tool–chip interface and reduce the effective contact area between the cutting tool and the chip. As a result, friction at the interface decreases, leading to improved surface quality. Similar lubricant-retention behaviour and friction reduction mechanisms have been reported for laser-textured carbide tools during machining of nickel-based superalloys [34,38]. Similar observations have been reported during machining of difficult-to-cut alloys using laser-textured carbide inserts where micro-textures enhanced lubricant retention and reduced tool–chip contact length [26,38].

An increase in cutting speed from 35 to 55 m/min resulted in a reduction in surface roughness for all tool configurations. This behaviour can be explained by the thermal softening of the work material at higher cutting speeds. The elevated temperature generated in the cutting zone reduces the shear strength of the material, allowing smoother chip formation and consequently improving the machined surface finish. Similar improvements in surface quality under

advanced lubrication environments have been reported during precision machining of superalloys using hybrid MQL-assisted cooling strategies [47].

Fig 4a illustrates the variation of surface roughness with feed rate (0.02, 0.04, and 0.06 mm/rev) at a cutting speed of 35 m/min. Similar trends were observed at the intermediate and highest cutting speeds. As the feed rate increases, surface roughness also increases due to the larger feed marks produced on the machined surface and the increased penetration of the cutting tool into the workpiece material.

Surface roughness increases with feed rate due to the ploughing action of the cutting tool on the workpiece surface. Similar trends have been observed for all the textured tools. Fig 4b showed the effects of varying cutting speed (35, 45 & 55 m/min) on the surface roughness with feed rate at 0.02 mm/rev; similar trends were observed for the feed rate of 0.04 & 0.06 mm/rev respectively.

### Effect of surface textures on cutting force (Fz)

The measured cutting force values under different machining conditions are presented in Table 4. Among the tested tools, the parallel-textured insert exhibited the lowest cutting force of 91.23 N under the seventh experimental condition. This observation agrees with the ANOVA summary (Table 3), where cutting speed was identified as the dominant parameter influencing cutting force. This value is lower than those recorded for the non-textured tool (101.38 N) and the other textured inserts (102.16 N for ET and 112.49 N for SCT). The reduction in cutting force is mainly attributed to improved

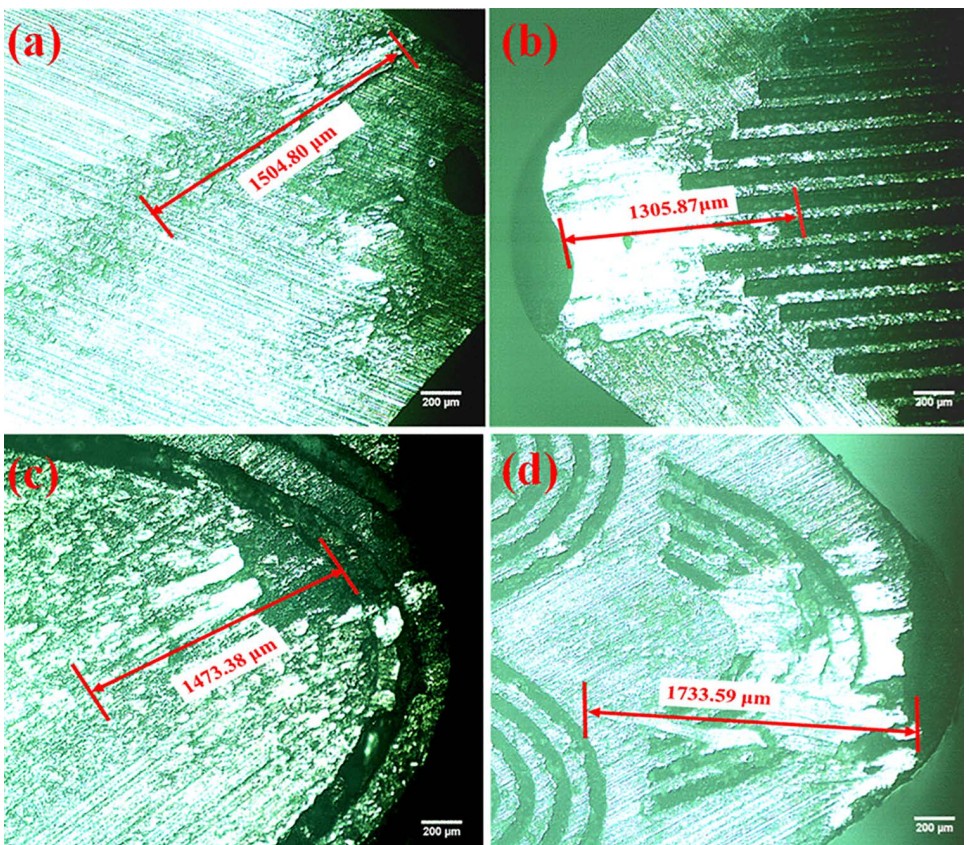

**Fig 4. Influence of cutting parameters on surface roughness (Ra): (a) variation with feed rate and (b) variation with cutting speed.**

lubrication and reduced tool–chip contact area provided by the parallel micro-textures. While cutting speed was increased from lowest (35) to the highest (55) m/min, cutting forces were reduced. It was due to thermal softening effect of the work piece (higher temperature in cutting zone). Fig 5a shows the effects of varying cutting speed with feed rate at 0.02, 0.04 & 0.06 mm/rev respectively. Cutting force increases with increasing feed rate due to more penetration of cutting tool into the work piece during machining. Similar trends have been observed for all the other textured tools considered for this experimentation [38]. Fig 5b shows the effects of varying feed rate on the cutting force with cutting speed at 35, 45 & 55 m/min respectively. Cutting force values decreased while cutting speed increases, it was due to thermal softening effect of the work piece as well as the effect of semi- solid lubricants also. The reduction in cutting force is due to better lubrication induces lower cutting temperature which eliminates detrimental effects of turning such as build up edge formation, adhesion and tool breakages. Similar trends have been observed for all the textured tools. It was noticed that the textured tools produced less cutting forces compared with that of non-textured tool as reported elsewhere [22,39,48].

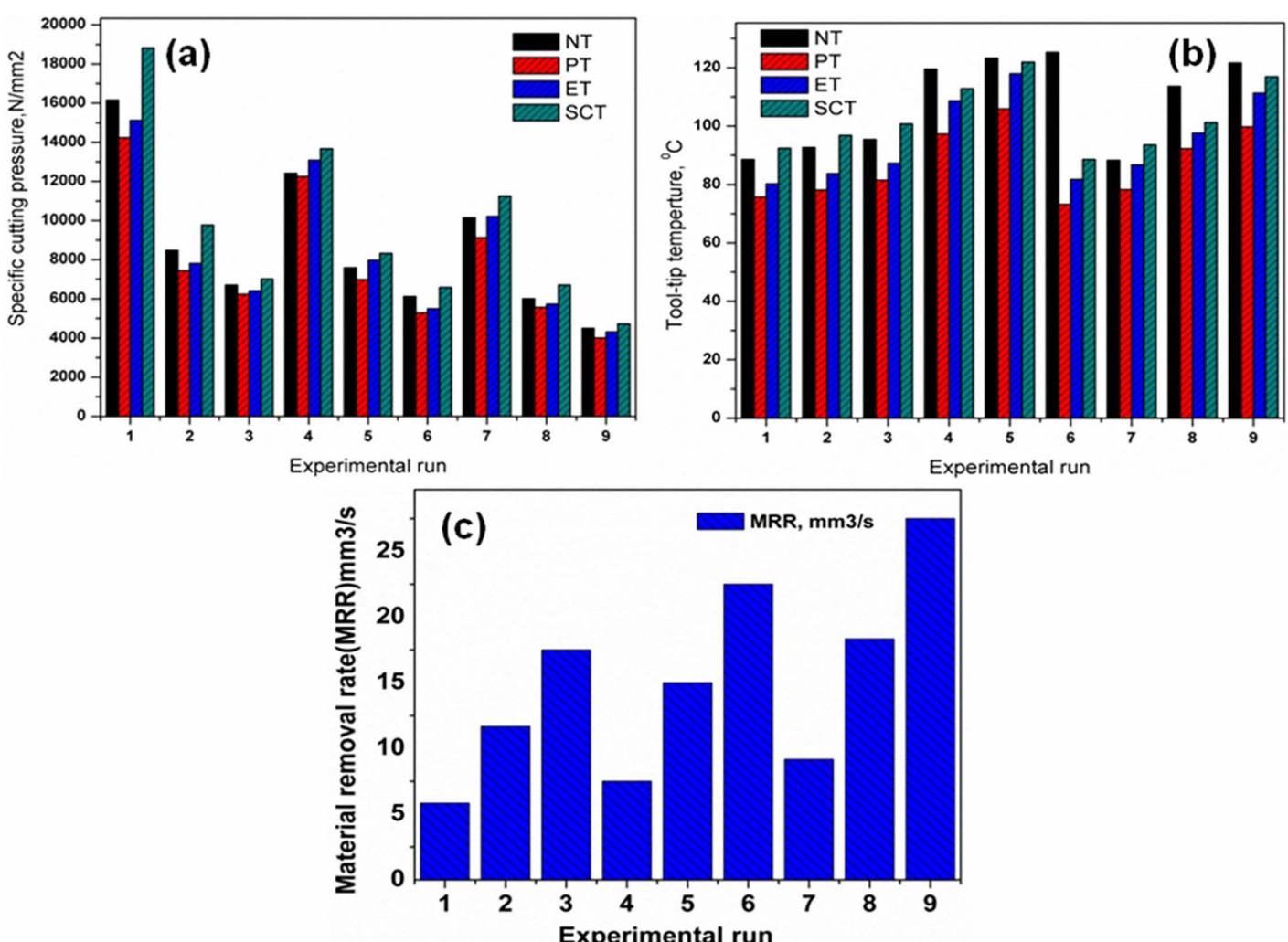

**Fig 5. Influence of machining parameters on cutting force (Fz) under different conditions: (a) variation of cutting force with feed rate (b) variation of cutting force with cutting speed.**

## Effect of surface textures on cutting power (P)

From the experimental results presented in Table 4 it is observed that PT- textured tool produced lowest cutting power 83.63W when compared with non – textured (NT; 92.93W) and other textured tools (ET; 93.65W & SCT; 103.12W). Fig 6a shows the effects of varying feed rate (0.02, 0.04 and 0.06 mm/rev) on the cutting power with cutting speed at 35 m/min. Fig 6b shows the effects of varying cutting speed on the cutting power with feed rate at 0.02 mm/rev.

When cutting speed is increased from 35 to 45 m/min, cutting power is increased due to strain hardening effect, from 45 to 55 m/min cutting power is reduced due to thermal softening effect. Generation of heat is enormous in cutting zone during turning operation and a reduction in shear strength of material by higher temperature, so that the required cutting force is less to deform the material at higher cutting speeds. Similar trends have been observed in cutting power with respect the higher levels of feed rate (0.04 and 0.06 mm/rev). It was noticed that the textured tools performed better compared with that of non-textured tool as reported elsewhere [22,25,48].

## Effect of surface textures on SEC

From the experimental results presented in Table 5, the highest specific energy consumption (16.17, 14.24, 15.13 & 18.14 J/mm$^3$ respectively for NT, PT, ET & SCT tool) obtained at the lowest levels of parameters (while cutting speed at 35 m/min & feed rate at 0.02 mm/rev) due to more strain-hardening effect and the lowest specific energy consumption (4.49, 4.00, 4.32 & 4.72 J/mm$^3$ respectively for NT, PT, ET & SCT tool) obtained at the highest levels of parameters (cutting speed at 55 m/min & feed rate at 0.06 mm/rev) it was due to the effect of thermal softening which is expected because of higher heat produced in the machining zone at higher cutting speed. Fig 7a shows the variations in SEC for varying cutting speed at 35, 45 & 55 m/min and Fig 7b shows the variation in SEC for different feed rate at 0.02, 0.04 & 0.06 mm/ rev. Similar trends have been observed for non-textured (NT) and other textured (ET & SCT) tools. The lowest SEC was observed for the PT textured tool (Experiment No. 9 in Table 4). The obtained results are in good agreement with the available literatures [24,25,38].

Comparison of machining performance indicators under different machining conditions for all textured and non textured tool inserts are shown in Fig 8. The surface roughness value variations for machined surfaces for the NT and different textured (ET & SCT) tools is shown in Fig 8a. Fig 8b shows the cutting force variations for the non-textured (NT) and other

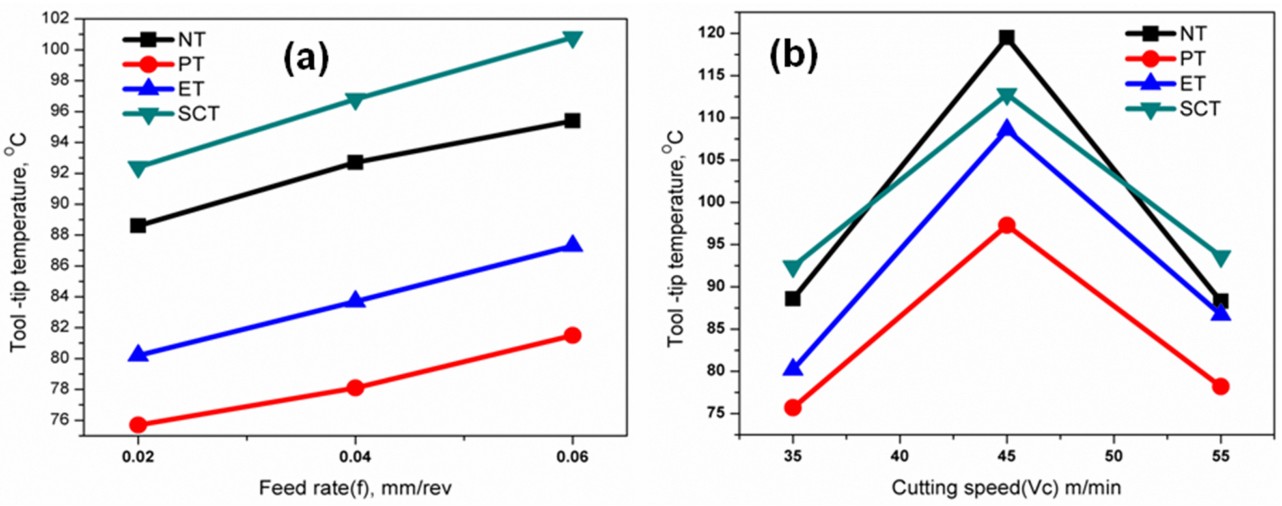

**Fig 6. Effect of machining parameters on cutting power under different lubrication conditions: (a) variation of cutting power with feed rate (b) variation of cutting power with cutting speed.**

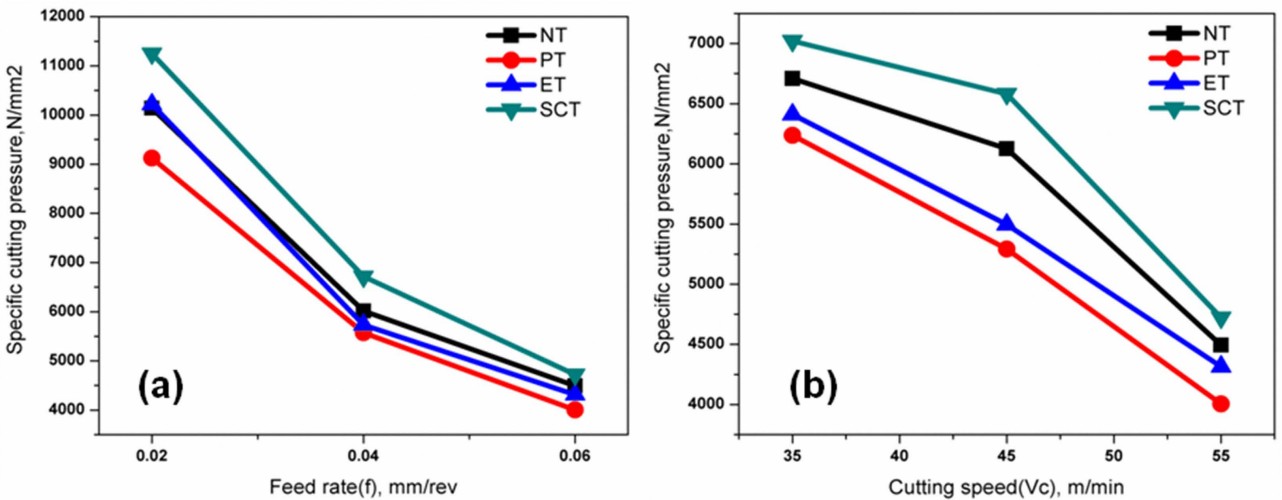

**Fig 7. Variation of specific energy consumption (SEC) under different conditions: (a) effect of cutting speed and (b) effect of feed rate.**

textured (ET & SCT) tools. Fig 8c shows the cutting power variations for the non-textured (NT) and other textured (ET & SCT) tools. Fig 8d shows the SEC variations for the non-textured (NT) and other textured (ET & SCT) tools. Comparing the non textured and other different textured tool inserts it is observed that the Parallel Textured tool inserts performed better for all the machining conditions.

## Effect of surface textures on SCPR

From the experimental results (Table 4) observed that the highest specific cutting pressure were 16162, 14236, 15123 & 18826 N/mm$^2$ respectively for NT, PT, ET & SCT tools; obtained at the lowest levels of parameters (cutting speed at 35 m/min & feed rate at 0.02 mm/rev) due to more strain hardening effect and the lowest specific cutting pressure (4493.33, 4004.66, 4315.66 & 4722.66 N/mm$^2$ respectively for NT, PT, ET &SCT tool) obtained at the highest levels of parameters (cutting speed at 55 m/min & feed rate at 0.06 mm/rev) due to thermal softening effect which is expected because of higher temperature produced in the cutting zone at higher cutting speed. Fig 9a showed the effects of varying feed rate on the SCPR with cutting speed at 35, 45 & 55 m/min respectively. Fig 9b showed the effects of varying cutting speed on the SCPR with feed rate at 0.02, 0.04 & 0.06 mm/rev respectively. The work material could be exposed to lesser amount of strain rate with the maximum feed rate during turning which lead to lower value of SCPR. In addition, the value of SCPR would start to decrease when increasing the cutting speed consequently reduction in shear strength which was due to higher value of temperature in the machining area. Similar trends have been observed for all the textured tools. The lowest SPCR was observed for the PT textured tool (9th experiment from Table 4).

## Effect of surface textures on tool-tip temperature

The cutting tool-tip temperature was measured taking 2 repeated trials at 2s interval at the various machining conditions. The measured tool-tip temperature at the tool–chip interface ranged from 88.3–125.3 °C for the NT tool, 73.2–105.9 °C for the PT tool, 80.2–117.9 °C for the ET tool, and 88.6–121.9 °C for the SCT tool. The statistical significance of cutting speed on tool-tip temperature variation is also confirmed by the ANOVA results summarized in Table 3. These temperatures correspond to the external rake-face region adjacent to the tool–chip interface measured using infrared sensing and therefore represent localized surface temperatures rather than the peak temperature within the primary shear deformation

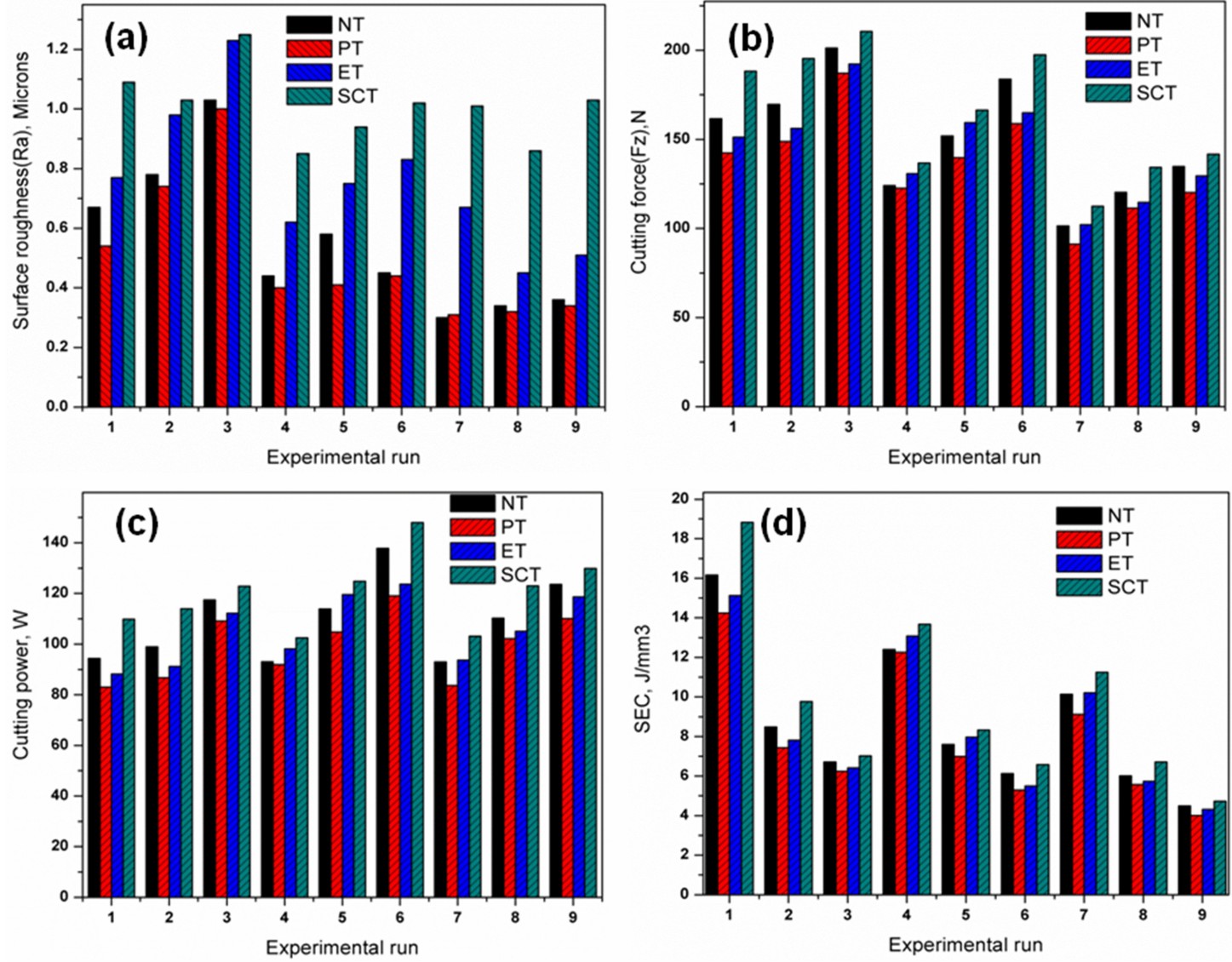

**Fig 8. Comparison of machining performance indicators under different lubrication conditions for all experimental runs: (a) surface roughness (Ra), (b) cutting force (Fz), (c) cutting power, and (d) specific energy consumption (SEC).**

zone, which is typically much higher during machining of nickel-based superalloys and may exceed 600 °C. Similar differences between infrared-measured surface temperature and actual shear-zone temperature have been reported in earlier machining studies of difficult-to-cut alloys [29]. While cutting speed was increased from 35 to 55 m/min, tool-tip temperature also were increased due to production of higher temperature in cutting zone during turning operation. However, textured surfaces with MoS$_2$ semi solid lubricant substantially enhanced the effective lubrication by means of a micro-pool lubrication mechanism with less tool–chip contact area that resulted in less frictional surfaces, hence, reduced cutting temperature. Fig 10a shows the effects of varying cutting speed on the tool-tip temperature with feed rate at 0.02, 0.04 & 0.06 mm/rev respectively. Fig 10b shows the effects of varying cutting speed on the tool-tip temperature with feed rate at 35, 45 & 55 m/min respectively. It was observed that PT textured tool produced lowest cutting tool-tip temperature

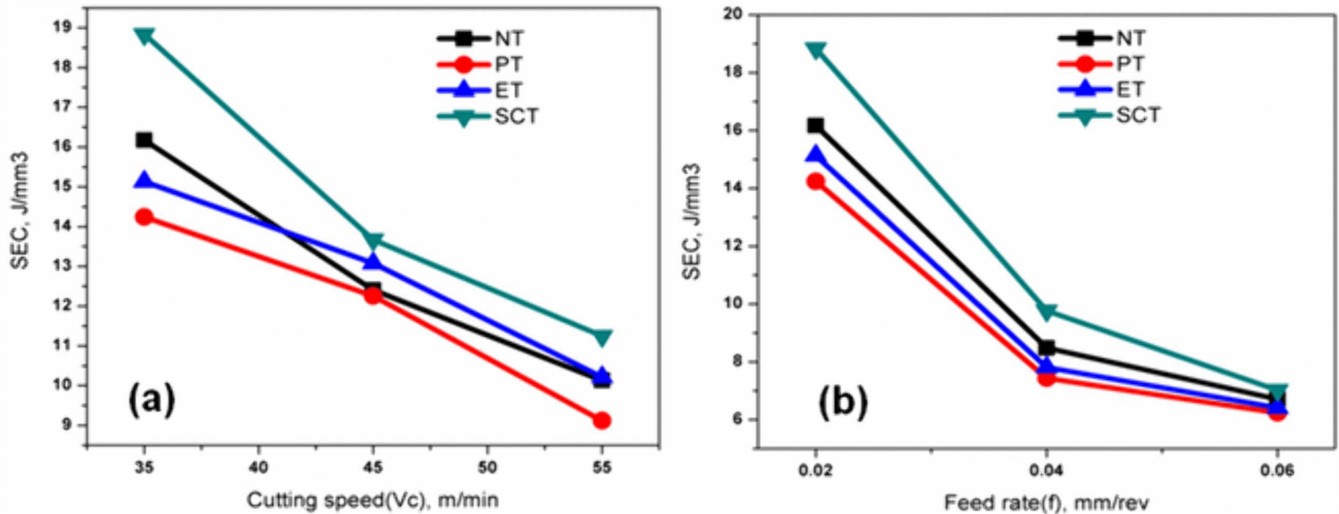

**Fig 9. Effect of machining parameters on specific cutting pressure under different conditions: (a) variation of specific cutting pressure with feed rate and (b) variation with cutting speed.**

compared with non – textured and other textured tools (ET & SCT). The slight reduction in temperature observed at higher cutting speed beyond 45 m/min may be attributed to reduced tool–chip contact time, thermal softening of the work material, and improved sliding behaviour promoted by semi-solid $MoS_2$ lubrication, which collectively decrease frictional heat generation despite increasing cutting velocity.

### Effect of surface texture on material removal rate (MRR)

It was noticed from Table 4, the highest MRR (27.50 mm³/s) was attained in ninth experiment with the cutting speed of 55 m/min, feed of 0.06 mm/rev and depth of cut of 1 mm. The MRR measured were ranges from 5.83 to 27.5 mm³/s. The MRR piercingly increases with increase of turning input variables from low level to high level [49,50]. It was caused the higher material removal from the substrate. The maximum MRR was obtained with highest cutting speed (55 m/min) due to the severe plastic-deformation and higher generation of heat (thermal softening effect of the work piece). Moreover, the generated heat induced by work-hardening effect for this alloy which controls the removal of material during turning operation. Fig 11a shows the SCPR variations for various textured tools. Fig 11b shows the Tool-tip temperature variations for various textured tools. Fig 11c shows the Material removal rate for different machining conditions. Comparing the non textured and other different textured tool inserts it is observed that the Parallel Textured tool inserts performed better in SCPR and Tool Tip temperature for all the machining conditions.

### Effect of surface texture on tool-chip contact length

Tool–chip contact length provides an indication of the frictional resistance encountered during chip flow over the rake face of the cutting tool. Lower friction reduces the sticking region between tool and chip, thereby decreasing the total contact length. Surface texturing improves the tribological behaviour of the tool by reducing friction and promoting lubricant retention at the interface. At higher cutting speed, generation of heat is higher at the tool-work piece interface causing thermal softening of the material at the region, reduces the chip contact length.

Consequently, lower frictional resistance is encountered by the tool during the machining. A higher contact length/area between tool and chip increases the friction and cutting temperature which lead to tool wear and poor surface quality.

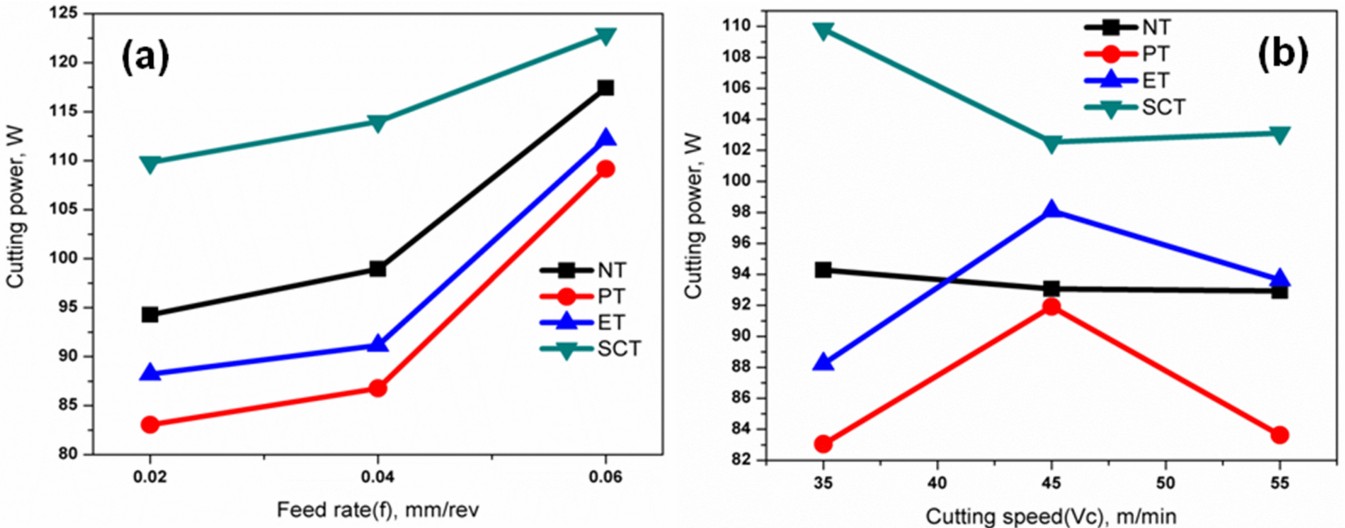

**Fig 10. Influence of machining parameters on tool-tip temperature under different conditions: (a) variation of tool-tip temperature with feed rate and (b) variation with cutting speed.**

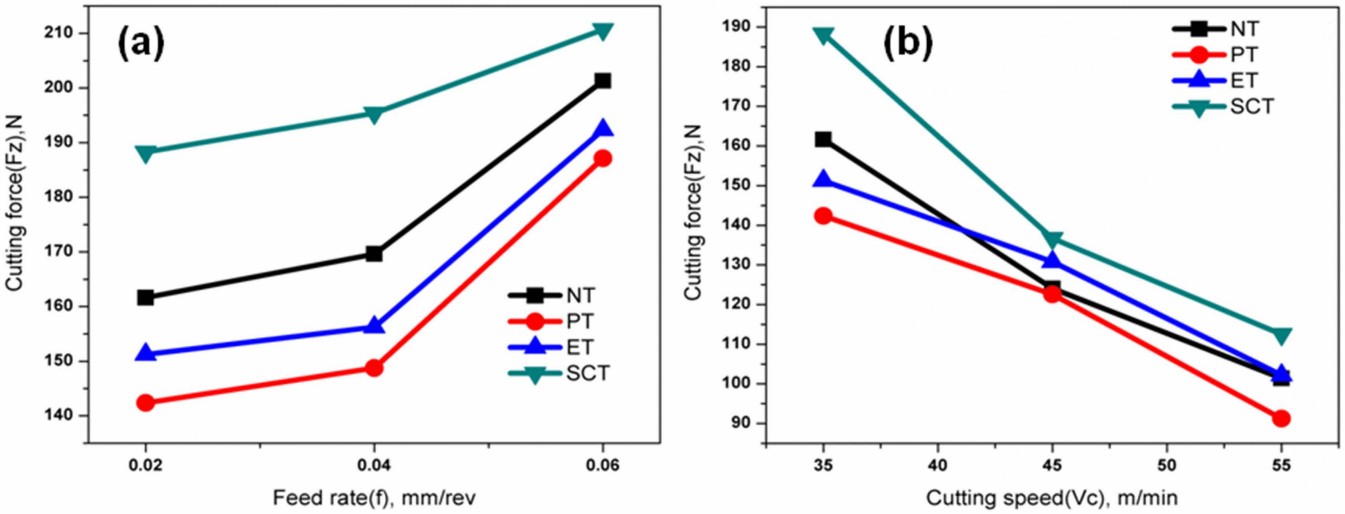

**Fig 11. Comparison of machining performance under different lubrication conditions across experimental runs: (a) specific cutting pressure, (b) tool–tip temperature, and (c) material removal rate (MRR).**

Reduction in tool–chip contact length was observed in PT tool compared with that of non-textured tool and ET and SCT tools (Fig 12b). Based on experimental results (Table 4), the PT textured tool having lowest tool-chip contact length (1305.87 µm) compared with NT, ET & SCT textured tools and forms longer tool-chip contact length over the tool (1504.80 µm, 1473.38 µm & 1733.59 µm) respectively and thus the power requirement is lowest for the same.

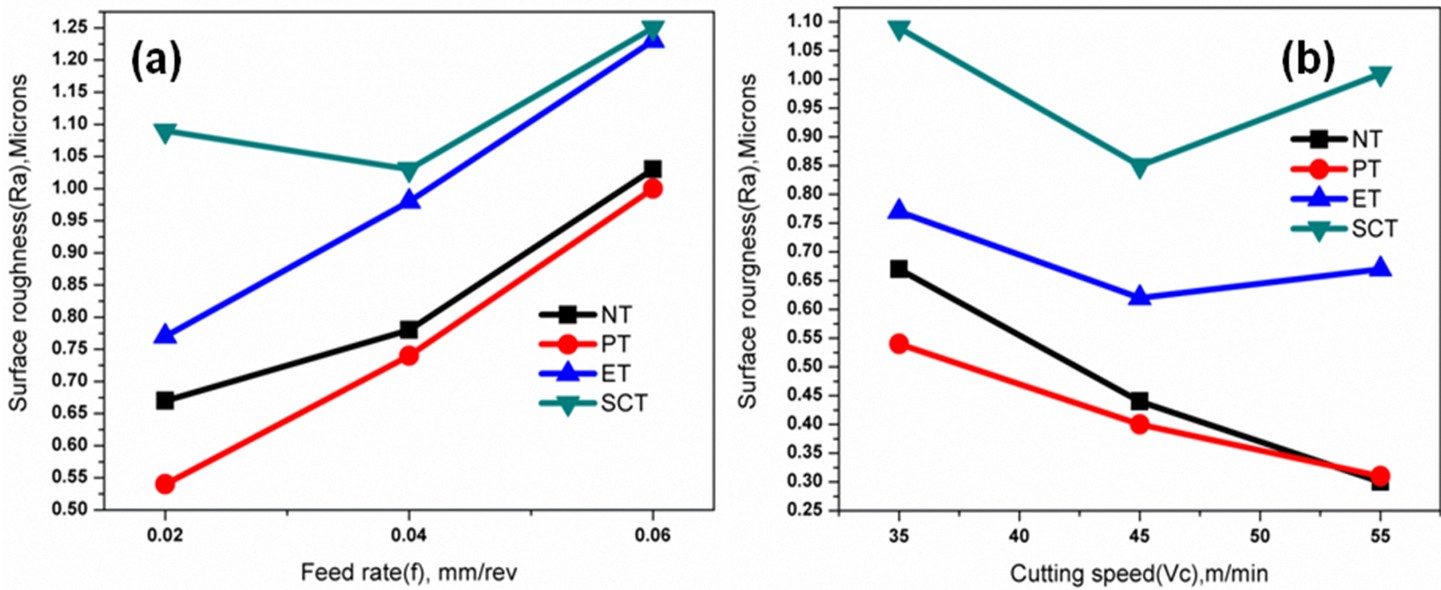

**Fig 12. Microscopic image of Tool-chip contact length for various textured tools (a) Non Textured, (b) Parallel Textured Tool, (c) Elliptical Textured Tool, (d) Semi Elliptical Textured Tool.**

## SEM and EDX analysis for the cutting tool wear

Tool wear is one of the most important indicators of machining performance, for that Scanning electron microscope (SEM, SU3500-Hitachi) with energy dispersive analysis (EDAX) was conducted in this work; The crater wear occurred in CNMA120408 NT tool after machining for the worst cutting condition, i.e., at cutting speed of 55 m/min, feed rate of 0.06 mm/rev and depth of cut of 1 mm (Ex.No.9) was shown in Fig 13a–c for the PT, ET and SCT tool respectively. It was noticed that here on the rake surface of the cutting tool, blended adhesive and diffusion mechanisms played a major role on the tool wear was shown in Fig 14a. It was attributed to high temperature induced between work material and tool rake face during machining.

Further, crater wear might have occurred by sliding friction caused by the chips over the rake face of the tool consequently some chips would start to stick over the rake face of the tool that promotes more friction. The formation of crater wear on the cutting insert is due to wear mechanisms of diffusion and abrasion. Diffusion played vital role in the creation of crater wear on the tungsten carbide insert at the tool-chip interfaces; because the amount of carbon in the tungsten carbide insert was significantly larger than that of Incoloy 800H. To elucidate diffusion at the tool-chip interfaces, corresponding spectrum of an EDAX analysis was done on the crater surface of the cutting insert which was shown in Fig 14b.

Correspondingly, Fe, Ni, Cr, Ti, Al, and Mg contents were good indicators for material transferred from the work material (Incoloy 800H) to the cutting insert (CNMA 120408 tungsten carbide tool) rake face during turning operations. Overall, the experimental results demonstrate that the orientation and geometry of laser-induced surface textures strongly influence machining behaviour. Parallel textures aligned with chip flow direction promote effective lubricant retention and reduce frictional resistance at the tool–chip interface. This mechanism results in lower cutting forces, reduced cutting temperature, and improved surface finish. Similar observations have been reported in previous studies on textured cutting tools used for machining difficult-to-cut alloys [39,48–50]. However, the present study extends these findings by demonstrating the effectiveness of semi-solid lubrication combined with texture geometry in machining Incoloy 800H. Although the present study provides valuable insights into the influence of laser surface texturing and semi-solid lubrication on machinability

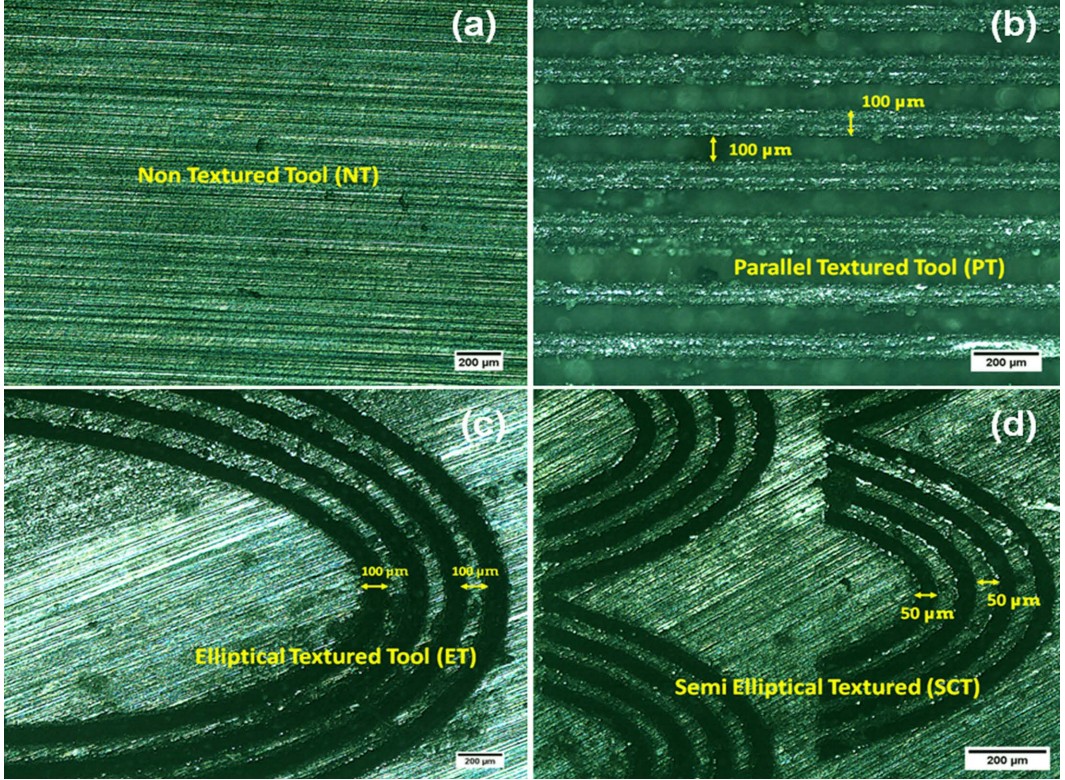

**Fig 13. (a-c) SEM images of the cutting inserts (PT, ET & SCT tool) after machining.**

of Incoloy 800H, the experiments were conducted under controlled laboratory conditions with limited cutting parameter ranges. Future studies may extend this investigation to higher cutting speeds, alternative tool materials, and industrial machining environments in order to further validate the applicability of the proposed approach. The observed improvement in machining performance for the PT textured tool agrees with previously reported tribological benefits of laser-textured cutting tools during machining of difficult-to-cut alloys.

## Conclusions

The machinability of heat-treated Incoloy 800H during CNC turning was experimentally investigated using laser-textured tungsten carbide inserts under semi-solid $MoS_2$ lubrication. The results demonstrate that the combined application of surface texturing and solid lubrication significantly improves machining performance by reducing frictional interaction at the tool–chip interface and enhancing lubrication retention during cutting. Among the tested configurations, the parallel-textured (PT) insert consistently exhibited superior performance compared with non-textured and other textured tools.

The major findings of this study are summarized as follows:

• Parallel-textured carbide inserts produced the lowest surface roughness (0.30 µm), representing approximately 12% improvement compared with non-textured inserts.

• Cutting force decreased by nearly 10% using PT inserts due to reduced tool–chip contact length and improved lubrication at the interface.

• Tool-tip temperature reduced by about 17% under semi-solid $MoS_2$ lubrication combined with surface texturing.

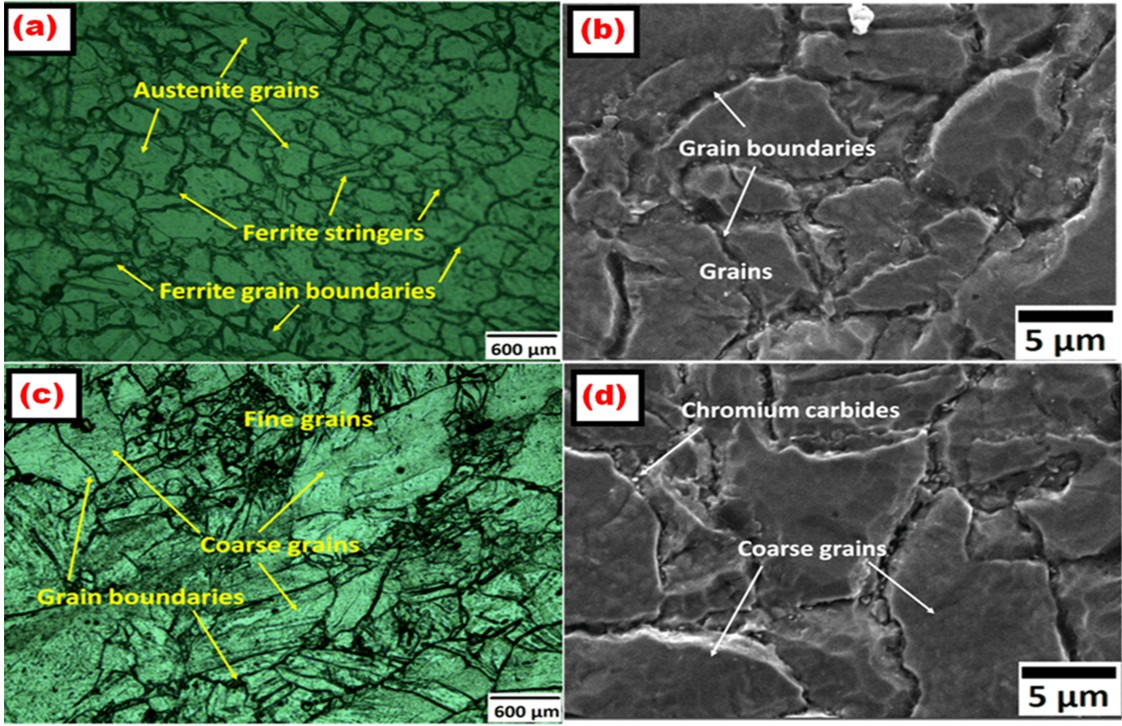

**Fig 14. (a) SEM image crater wear analysis of CNMA120408 NT tool (b) Corresponding EDAX analysis on tool tip of CNMA 120408 NT tool.**

- Specific energy consumption decreased by approximately 11%, confirming improved machining efficiency during turning.

- The PT insert produced the lowest specific cutting pressure (4004.6 N/mm²), indicating reduced resistance to material deformation.

- Tool–chip contact length reduced from 1504.80 µm to 1305.87 µm, demonstrating reduced frictional interaction at the cutting interface.

- SEM–EDX analysis confirmed diffusion and abrasion as the dominant tool-wear mechanisms during machining of heat-treated Incoloy 800H

Overall, the results confirm that laser surface texturing combined with semi-solid $MoS_2$ lubrication provides an effective and practical strategy for improving the machinability of heat-treated Incoloy 800H under controlled turning conditions, particularly when parallel-oriented textures are aligned with chip flow direction.

## Supporting information

**S1 Data. Raw experimental dataset used for statistical evaluation and generation of all plots reported in this study.** (PDF)

## Acknowledgments

The authors would like to thank Surya Engineering College, Erode, India; Adhi College of Engineering and Technology, Kancheepuram, India; Roever Engineering College, Perambalur, India; and K. Ramakrishnan College of Engineering, Tiruchirappalli, India for providing institutional support during the course of this research. The authors also acknowledge

the College of Science and Technology, Royal University of Bhutan, Phuentsholing, Bhutan for experimental, academic support and collaboration. The authors gratefully acknowledge Test Point Research Laboratory, Coimbatore, India for carrying out the spectroscopy analysis of the work material. The authors also thank Meeras Laser Solutions, Chennai, India for their technical assistance in fabricating the laser surface textures on the cutting inserts.

## Author contributions

**Conceptualization:** Palanisamy Angappan, Palanisamy Duraisamy, Prakash Chellamuthu, Abhishek Agarwal, Lenin Kasirajan.

**Data curation:** Palanisamy Angappan, Palanisamy Duraisamy, Prakash Chellamuthu, Abhishek Agarwal, Lenin Kasirajan.

**Formal analysis:** Palanisamy Angappan, Palanisamy Duraisamy, Prakash Chellamuthu, Lenin Kasirajan.

**Funding acquisition:** Abhishek Agarwal.

**Investigation:** Prakash Chellamuthu, Lenin Kasirajan.

**Methodology:** Palanisamy Angappan, Palanisamy Duraisamy, Prakash Chellamuthu, Abhishek Agarwal, Lenin Kasirajan.

**Project administration:** Palanisamy Duraisamy, Abhishek Agarwal.

**Resources:** Palanisamy Angappan, Palanisamy Duraisamy, Abhishek Agarwal, Lenin Kasirajan.

**Software:** Palanisamy Angappan, Palanisamy Duraisamy, Prakash Chellamuthu, Abhishek Agarwal.

**Supervision:** Abhishek Agarwal.

**Validation:** Palanisamy Angappan, Palanisamy Duraisamy, Prakash Chellamuthu, Abhishek Agarwal, Lenin Kasirajan.

**Visualization:** Palanisamy Angappan, Prakash Chellamuthu, Abhishek Agarwal, Lenin Kasirajan.

**Writing – original draft:** Palanisamy Angappan, Palanisamy Duraisamy, Prakash Chellamuthu, Abhishek Agarwal, Lenin Kasirajan.

**Writing – review & editing:** Palanisamy Angappan, Palanisamy Duraisamy, Prakash Chellamuthu, Abhishek Agarwal, Lenin Kasirajan.

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
