## [Decision Letter · Decision Letter 0]

12 Apr 2026

PONE-D-26-12246Machinability enhancement of heat-treated Incoloy 800H during turning using laser-textured carbide inserts under semi-solid MoS2 lubricationPLOS One

Dear Dr. Agarwal,

Thank you for submitting your manuscript to PLOS ONE. After careful consideration, we feel that it has merit but does not fully meet PLOS ONE’s publication criteria as it currently stands. Therefore, we invite you to submit a revised version of the manuscript that addresses the points raised during the review process. Please submit your revised manuscript by May 27 2026 11:59PM. If you will need more time than this to complete your revisions, please reply to this message or contact the journal office at plosone@plos.org. Please include the following items when submitting your revised manuscript:

We look forward to receiving your revised manuscript.

Kind regards,

Mithilesh K. Dikshit

Academic Editor

PLOS One

Journal Requirements:

2. We note that your Data Availability Statement is currently as follows: “All relevant data are within the manuscript and its Supporting Information files.”

3. We notice that your supplementary figures are uploaded with the file type 'Figure'. Please amend the file type to 'Supporting Information'. Please ensure that each Supporting Information file has a legend listed in the manuscript after the references list.

Additional Editor Comments:

The authors must incorporate the suggestions provided by the reviewers in the revised manuscript.

Reviewer's Responses to Questions

**Comments to the Author**

1. Is the manuscript technically sound, and do the data support the conclusions?

Reviewer #1: Partly

Reviewer #2: Yes

Reviewer #3: Yes

2. Has the statistical analysis been performed appropriately and rigorously? 

Reviewer #1: No

Reviewer #2: Yes

Reviewer #3: No

3. Have the authors made all data underlying the findings in their manuscript fully available?

Reviewer #1: Yes

Reviewer #2: Yes

Reviewer #3: Yes

4. Is the manuscript presented in an intelligible fashion and written in standard English?

Reviewer #1: Yes

Reviewer #2: Yes

Reviewer #3: Yes

5. Review Comments to the Author

Reviewer #1: 1) Include more recent articles in the introduction section

2) The objective and novelty in the research should be included in the last paragraph of the introduction section.

3) Include the laser texture parameters selected

4) Include the dimensions of the textured surfaces on the rake surface.

5) Results and discussion section- This improvement can be attributed to the orientation of the parallel micro-textures, which promote lubricant retention at the tool–chip interface and reduce the effective contact area between the cutting tool and the chip. As a result, friction at the interface decreases, leading to improved surface quality – include citation to validate this statement

6) Generally nickel based alloys has poor thermal conductivity. Hence, the highest temperature recorded were 125°C. It is not possible. Check this

7) Results and discussion section- include more citations to validate your findings.

8) Rewrite the conclusion in a point wise.

9) Conclusion – include the % wise comparison of your findings.

Reviewer #2: This study experimentally investigates the machinability of heat

treated Incoloy 800H during CNC turning using laser-textured tungsten carbide cutting inserts

combined with semi-solid MoS₂ lubrication. The findings demonstrated that the combined application of laser

41 surface texturing and semi-solid MoS₂ lubrication significantly improves the machinability of

42 Incoloy 800H during turning operations.

The article is very interesting and relevant.

Forged rods of Incoloy 800H superalloy with a diameter of 30 mm and length of 120 mm were

used as the work material. how dimesion is fixed?

- Table 1. Chemical composition of Incoloy 800H (wt %). provide citation.

- Table 2. Experimental design layout for the turning experiments indicating cutting speed (Vc),

feed rate (f), and depth of cut (ap). how the parameters are selected? Why Depth of cut (ap),

(mm) is constant ? Which design of experiment has been adopted?

- How texturing ids done? Conclusions may be stated point wise.

Refer following papers :

Optimizing Sustainable Machining Conditions for Incoloy 800HT Using Twin-Nozzle MQL with Bio-Based Groundnut Oil Lubrication

Sustainable Precision Machining of Superalloy Using Hybrid MQL-N2 Cooling: Improving Surface Quality and Dimensional Accuracy

Performance improvement through textured cutting tool during machining hard-to-cut materials: a review

Reviewer #3: The article entitled "Machinability enhancement of heat-treated Incoloy 800H during turning using laser-textured carbide inserts under semi-solid MoS2 lubrication" deals with an important problem regarding the machinability and quality of parts during the turning of nickel alloys. The title corresponds to the content of the article. I believe that the article can be published in the PLOS One after considering the following suggestions:

How often were the individual cutting tests repeated? Are the values shown average or maximum? What was the dispersion of values?

What was the depth of the textures? After how long of cutting time did the cutting tool wear greater than the depth of the applied texture? At what distance from the cutting edge was the texture of PT (Fig. 3a), and was it in the range of cutting depth ap=1mm?

MoS2 lubrication description should be extended.

The description of the temperature measurements requires further clarification – how exactly were the measurements performed? Were the tool temperature measured, or the temperature of the chip formed on the rake face?

Fig. 10 b – the authors should explain the reason for the increase and then decrease in temperature for vc=45 m/min and vc=55 m/min, respectively.

The authors did not present a statistical analysis of the results of the experimental research, as well as a mathematical model characterising the influence of the analysed factors on the measured parameters. In addition, there is a lack of analysis of the significance of the impact of individual factors on the measured values. In the individual chapters describing the results obtained, the authors should also present the explanations of the observed phenomena in more detail and to a greater extent, referring to the physical phenomena occurring in the cutting zone.

6. PLOS authors have the option to publish the peer review history of their article (what does this mean?). If published, this will include your full peer review and any attached files.

Reviewer #1: No

Reviewer #2: No

Reviewer #3: No

---

## [Author Response · Author response to Decision Letter 1]

13 Apr 2026

We thank the Academic Editor and the Reviewers for their careful evaluation and constructive comments on our manuscript. All reviewer and editor suggestions have been addressed in detail in the revised manuscript. A point-by-point response to each comment has been provided in the uploaded “Response to Reviewers” document, and all corresponding revisions have been incorporated in the tracked-changes version of the manuscript.

---

## [Decision Letter · Decision Letter 1]

26 Apr 2026

Machinability enhancement of heat-treated Incoloy 800H during turning using laser-textured carbide inserts under semi-solid MoS2 lubrication

PONE-D-26-12246R1

Dear Dr. Agarwal,

We’re pleased to inform you that your manuscript has been judged scientifically suitable for publication and will be formally accepted for publication once it meets all outstanding technical requirements.

Kind regards,

Mithilesh K. Dikshit

Academic Editor

PLOS One

Additional Editor Comments (optional):

The manuscript is accepted in the present form.

Reviewers' comments:

Reviewer's Responses to Questions

**Comments to the Author**

1. If the authors have adequately addressed your comments raised in a previous round of review and you feel that this manuscript is now acceptable for publication, you may indicate that here to bypass the “Comments to the Author” section, enter your conflict of interest statement in the “Confidential to Editor” section, and submit your "Accept" recommendation.

Reviewer #1: All comments have been addressed

Reviewer #2: All comments have been addressed

Reviewer #3: All comments have been addressed

2. Is the manuscript technically sound, and do the data support the conclusions?

Reviewer #1: Partly

Reviewer #2: Yes

Reviewer #3: Yes

3. Has the statistical analysis been performed appropriately and rigorously? 

Reviewer #1: Yes

Reviewer #2: Yes

Reviewer #3: Yes

4. Have the authors made all data underlying the findings in their manuscript fully available?

Reviewer #1: Yes

Reviewer #2: Yes

Reviewer #3: Yes

5. Is the manuscript presented in an intelligible fashion and written in standard English?

Reviewer #1: Yes

Reviewer #2: Yes

Reviewer #3: Yes

6. Review Comments to the Author

Reviewer #1: the authors had addressed all the queries raised by the reviewers. Hence it can be accepted in its present form

Reviewer #2: The manuscript has undergone substantial revision and shows clear, measurable improvement. The authors have addressed the reviewers’ main concerns by clarifying the research article's central contributions, strengthening the methodological description, and providing additional analyses that support the main finding.

The paper now accepted and recommended for publications.

Reviewer #3: I thank the authors for their responses. The explanations are sufficient and adequate. I accept additions.

7. PLOS authors have the option to publish the peer review history of their article (what does this mean?). If published, this will include your full peer review and any attached files.

Reviewer #1: No

Reviewer #2: No

Reviewer #3: No

---

## [Editor Report · Acceptance letter]

PONE-D-26-12246R1

PLOS One

Dear Dr. Agarwal,

I'm pleased to inform you that your manuscript has been deemed suitable for publication in PLOS One. Congratulations! Your manuscript is now being handed over to our production team.

Kind regards,

on behalf of

Dr. Mithilesh K. Dikshit

Academic Editor

PLOS One